# Impact of solvent forces and broken symmetry on the assembly of designed proteins at a liquid-solid interface

Sakshi Yadav Schmid[1,2,11,12,17], Benjamin Helfrecht [1,17], Amy Stegmann[3,17], Benjamin A. Legg [1], Harley Pyles[4,5], Jiajun Chen[2,13], John R. Edison[6,14], Maxim Ziatdinov[1,7], Zdenek Preisler[6,15], Orion Dollar[8,16], Stephen Whitelam [6], Sergei Kalinin [9], David Baker [4,5,10], Christopher J. Mundy [1,8] ✉, Shuai Zhang [1,2] ✉ & James J. De Yoreo [1,2] ✉

The era of protein design has enabled the creation of hybrid protein-inorganic interfaces, leading to both surface-directed self-assembly of de novo protein architectures and protein-directed formation of inorganic materials. However, the resulting patterns of protein assembly are often unexpected, implying that essential interactions are not accounted for in current design platforms. Here, we use high-speed atomic force microscopy (AFM) analyzed through machine learning to follow the assembly of protein nanorods in aqueous electrolytes on two types of mica exhibiting disparate symmetry elements, which are imprinted on the overlying hydration structure. Using Monte Carlo simulations, we reproduce the observed phases and show that an observed smectic phase, previously thought to be unstable for non-interacting rods in two dimensions, emerges when crystal symmetry introduces a directional bias. The findings demonstrate the importance of incorporating solvent forces as modulated by the hydration structure inherent to interfacial systems when designing protein assemblies at liquid-crystal interfaces. Coupling physics-based simulations that can account for these factors to de novo protein design algorithms can lead to improved design platforms for bio-inspired, hybrid materials.

Nature's ability to create hybrid biomolecular-inorganic materials with a hierarchy of structural motifs has inspired extensive research into the assembly of macromolecules at solid surfaces[1–10]. Efforts to intentionally design proteins, peptides, and peptide mimetics to self-assemble at surfaces and exhibit order across multiple length scales have led to promising progress with potential applications in energy harvesting, catalysis, photonics, biomedicine, and structural materials[11–13].

Strategies to direct protein assembly have generally focused on manipulating chemical bonds between the proteins, electrostatic interactions between charged functional sidechains and surfaces, and hydrophobic interactions both between the proteins themselves and

with the surface[1,8,12,13]. However, even when a set of interactions has been intentionally designed into the sequence, phases emerge that do not represent the design point. For example, when the enzyme RhuA was site-specifically modified to form two-dimensional (2D) lattices through in-plane chemical bonds, assembly in bulk solution produced the target structure, but assembly on mica surfaces led to three distinct and unexpected monolayer and bilayer phases[3]. In previous research studies using de novo design to create protein nanorods exhibiting a precise match to the crystal lattice of mica, a rich set of ordered 2D structures was observed, but many were not design targets and could not have been anticipated from lattice matching alone[1].

---

The common occurrence of unexpected phases demonstrates that the designed interactions do not fully encompass the suite of forces that drive protein assembly at surfaces and cannot be used alone to predict the outcome. Thus, computational protein design platforms that design the tertiary structures of protein molecules and quaternary structures of protein assemblies cannot yet be exploited to predictively design assemblies of proteins on inorganic surfaces. Given that proteins, despite their atomic precision, are akin to electrostatically patchy particles interacting with both the surface and the surrounding solution, the role of colloidal forces and the impact of the interfacial solution structure, both of which are modulated by solution electrolytes and the broken symmetry imposed by the substrate[14–19], must be considered but are not currently an element of these design platforms.

Here, we explore the impact of inherent interfacial forces on the off-target assembly of designed proteins at a liquid-mineral interface. We use machine learning to analyze in situ high-speed atomic force microscopy (HS-AFM) data and quantify the assembly dynamics and degree of order achieved by protein nanorods during assembly on surfaces to which they are lattice-matched. We additionally employ Monte Carlo simulations to predict the dependence of order on chemical potential, mobility, and symmetry of the potential energy landscape. The results show that the consideration of these interfacial forces leads to an accurate prediction of the observed 2D phases. In particular, the emergence of a 2D smectic phase, which is unexpected based on long-standing theories of colloidal systems[20,21] as well as

simulations[22,23], is shown to arise naturally when symmetry breaking creates a two-fold bias in the potential energy landscape. The results suggest a method for integrating coarse-grain simulations of colloidal systems into de novo protein design platforms, enabling the accurate design of supramolecular protein architectures at inorganic interfaces.

## Results

### Assembly of DHR protein on mica

To investigate the role of interfacial structure on protein assembly, we utilized a rectangular rod-shaped de novo designed helical repeat (DHR) protein[1] with an aspect ratio of 1:5.6− referred to as DHR10-mica18. Using the Rosetta de novo protein design platform, we previously designed this protein to interact with the (001) cleavage plane of mica by creating a protein scaffold having a flat surface and a regularly repeating backbone with a repeat spacing equal to an integer multiple of the 5.2 Å lattice spacing between nearest-neighbor $K^+$ sites, which form a hexagonal sublattice (Fig. 1a, b)[1]. As indicated by the suffix "18", this protein consists of 18 repeating subunits, each with three glutamate residues. Together they form an array of 54 carboxylate groups positioned to exhibit a structural match to the $K^+$ sublattice (Supplementary Fig. 1). Due to this match, the proteins are expected to bind electrostatically to the $K^+$ sites along the three equivalent directions (Fig. 1c, g). For details of the protein sequence, synthesis, and characterization, see ref. 1. While DHR10-mica binding proteins with different numbers of repeats were explored in our previous research[1,24], we chose the 18-repeat version for this study because

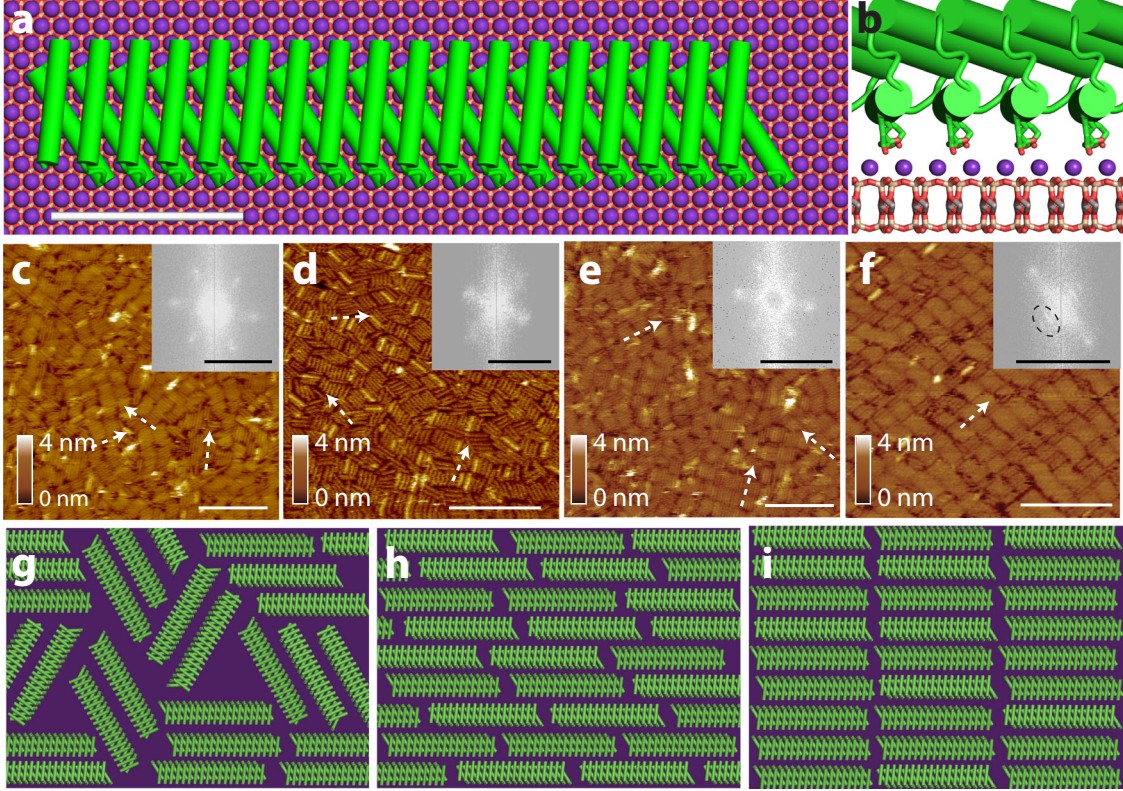

**Fig. 1 | Protein monomer design lattice matched to the mica surface and assembly outcome. a** The Rosetta design model of a DHR10-mica18 protein nanorod with dimensions 3.6 nm × 20 nm adsorbed on mica surface. The protein consists of 18 tandem repeat units shown in green with alpha-helices rendered as cylinders. An aluminosilicate layer of the mica substrate is shown with the $K^+$ sublattice (shown as purple spheres). Scale bar is 5 nm. **b** Side view of the protein-mica interface showing negatively charged glutamate side chains (green and red sticks, respectively) extending from the protein with a periodicity that forms a 2-to-1 lattice match with the mica surface. **c, d** AFM images of the final assembly states of

DHR-mica18 on f-mica and m-mica in 100 mM and **e, f** 3 M KCl, respectively. Scale bars are 100 nm. The FFT is shown in the inset. The FFT scale bars are 0.5 nm⁻¹. Note that the observed phases are observed both during in situ imaging and after extraction from solution. In addition, the same orientation of the rods in (f) is observed everywhere across the surface. **g–i** Illustrations of 2D phases of hard rods possible at high concentrations in a three-fold potential: **g** 2D high-density disordered (HDD) phase, **h** nematic phase, and **i** smectic phase. Note that the smectic phase is not predicted when the rods are non-interacting[21].

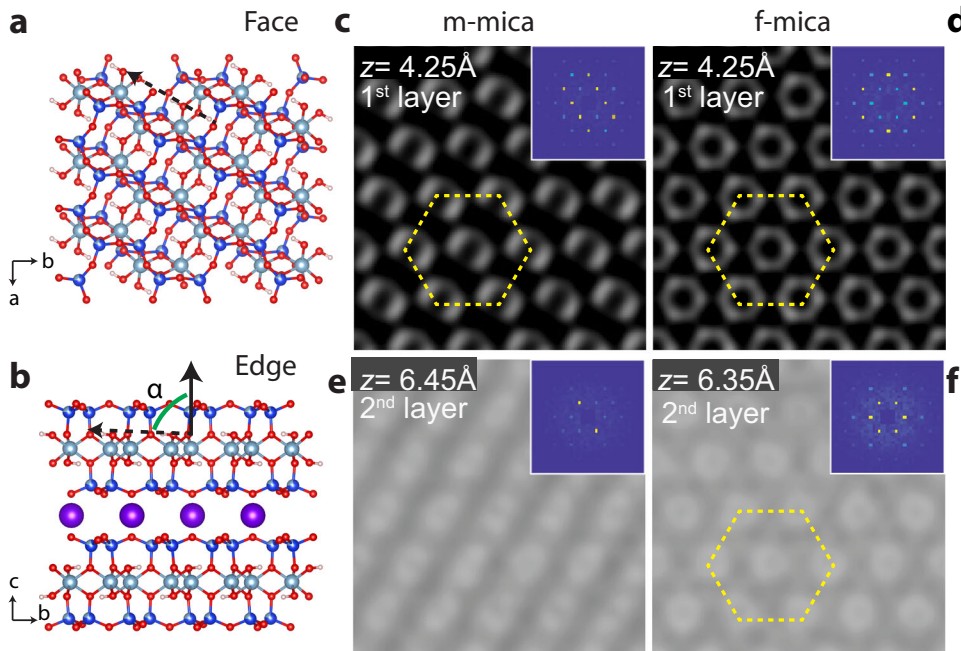

**Fig. 2 | Structure of mica and overlying hydration layers. a, b** Atomic models showing face and edge views of two m-mica layers. The face (**a**) shows both the hexagonal array of cavities within a tetrahedrally coordinated aluminosilicate sheet in which K$^+$ ions sit, as well as the hydroxyl groups that lie below the cavities and point alternately along one of two axes of the K+ sublattice, but not along the third ([100] axis). The edge (**b**) view shows that these hydroxyl groups lie in each successive layers below the surface. In addition, below the surface layer, a partially occupied layer of octahedrally coordinated Al$^{3+}$ ions combines with the hydroxyls to break the three-fold symmetry and slightly distort the aluminosilicate tetrahedral sheet. In f-mica, the octahedrally coordinate layer is fully occupied by Mg$^{2+}$ ions and the hydroxyl groups are replaced by F- atoms, rendering the structure three-fold symmetric (see Supplementary Fig. 2 for details). Purple: potassium, blue: silicon, cyan: aluminum, red: oxygen, white: hydrogen. A wedge of cyan in the tetrahedral layer represents the partial substitution of silicon with aluminum. **c–f** Oxygen density in the first and second hydration layers, as predicted by molecular dynamics simulations, reveals the hexagonal symmetry of both layers on m-mica and f-mica, and the emergence of a striped pattern in the second layer above m-mica, resulting from the broken symmetry[18]. See Supplementary Figs. 2 and 3 for further details, including three-dimensional AFM measurements validating the predictions. **c–f** Reproduced with permission from ref. 15 (copyright American Chemical Society 2022).

its aspect ratio was sufficiently large to ensure a significant entropic force for co-alignment while small enough for the proteins to be relatively rigid, thus approximating hard rods.

We chose two types of mica for the study − muscovite (m-) and fluorophlogopite (f-) mica −because previous results showed that, under conditions of high KCl concentration, both DHR10-mica proteins and others exhibit differences in their organization on the two mica types, despite having identical K$^+$ sublattices[1,3]. In addition, the structural differences between m- and f-mica[25–27], which are unrelated to the K$^+$ sublattice, were shown to create distinct differences in the structure of the overlying hydration layers (Fig. 2 and Supplementary Figs. 2, 3)[15], with both the first and second hydration layers above f-mica predicted and observed to exhibit hexagonal symmetry while the broken symmetry of the m-mica lattice leads to the emergence of a striped pattern in the second hydration layer (Supplementary Fig. 3)[15]. For experimental protocols, see Methods. Briefly, the protein stock solution was diluted to the desired concentration with the incubation buffer containing 20 mM Tris-HCl (pH 7) and 3 M KCl. Then, 10 μL of diluted protein solution was dropped onto freshly cleaved substrates at room temperature and the imaging was started immediately to capture the assembly process in situ, under constant ambient conditions.

In agreement with ref. 1, we find that, in 100 mM K$^+$ aqueous solutions, the protein nanorods assemble into a disordered phase consisting of small domains of coaligned proteins oriented along the three principal axes of the underlying mica lattice, regardless of whether f- or m-mica is used (Fig. 1c, d). The 2D fast Fourier transform (FFT) of Fig. 1c-e all display three pairs of blurry high-intensity spots. These represent the short-range order of the co-aligned protein nanorods within small domains, but they lack long-range order. However, when

the K$^+$ concentration is increased to 3 M, DHR10-mica18 continues to form the three-fold disordered phase on f-mica (Fig. 1e), but it assembles into an ordered phase on m-mica in which all rods are coaligned along a single direction corresponding to the unique axis of m-mica and arranged in parallel rows everywhere across the m-mica substrate (Fig. 1f). Additionally, the corresponding 2D FFT shows two sets of condensed spots, representing the side-by-side arrangement of the co-aligned protein nanorods in a row and the single directional long-range order of the co-aligned nanorods across the rows[1].

In the parlance of the liquid crystal literature, the observed disordered phase is known as a 2D high-density disordered (HDD) phase (Fig. 1g) and is predicted for a sufficiently high rod concentration in a 3-fold potential when the translational and rotational mobility are low[22,23]. As the mobility increases, theoretical treatments predict that the rods will align due to purely entropic forces, forming a 2D nematic phase[21] (Fig. 1h), which we do not observe experimentally. Instead, the ordered phase observed on m-mica at 3 M K$^+$ has smectic order, (Fig. 1i) which is not predicted for non-interacting rectangular rods in 2D, though the introduction of excluded volume interactions at the rod tips due to the addition of polymer tails or charges that produce electrostatic repulsion has been predicted to stabilize smectic order.

The above observations thus present a conundrum: if the rod-substrate potential is established by the interaction of the glutamate side chains with the K$^+$ ions of the mica lattice, then why should the nanorods assemble into two distinct phases on m- and f-mica, which possess identical K$^+$ sublattices, and, for the case of m-mica, why does smectic order emerge in a two-dimensional system, rather than only reaching nematic order? To answer these questions, we used HS-AFM[4] to observe protein adsorption and assembly at the water-mica

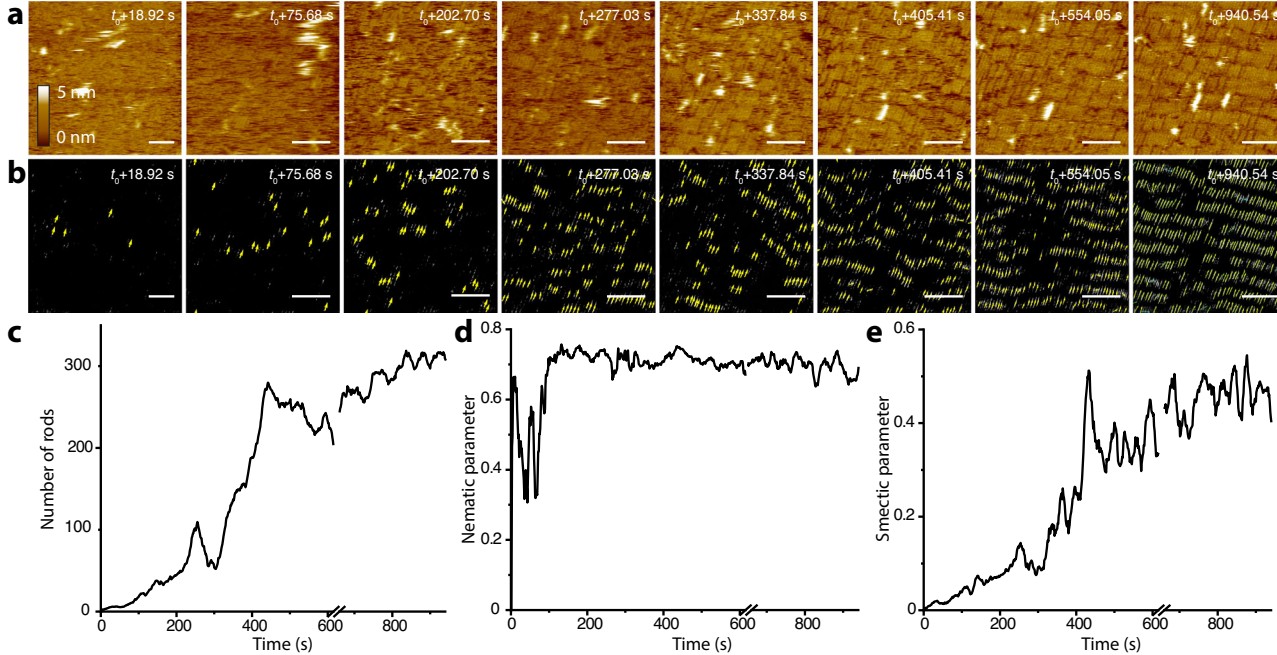

**Fig. 3 | In-situ high-speed AFM results and machine learning analysis to follow the assembly of protein nanorods on m-mica. a** HS-AFM images from Supplementary Movie 1 showing the translational motion of protein rods and their assembly on m-mica into a smectic phase. Scale bar equals 50 nm. **b** Machine-learning-based workflow to recognize the protein rods in the HS-AFM images in (**a**), where each arrow indicates the placement and alignment of a nanorod. **c** Number of recognized protein rods as a function of time. **d, e** Nematic and smectic order parameters as a function of time, respectively, for the observed rod assembly in (**a**). The discontinuity in the data lines in (**a–c**) is due to the in situ HS-AFM losing track during the experiment. Source data are provided as a Source Data file.

interface, follow the emergence of order, and quantify the degree of nematic and smectic order on both substrates.

In the case of m-mica in 3 M K⁺ (Fig. 3 and Supplementary Movie 1), individual proteins are rarely observable in the initial frames; rather DHR10-mica18 initially forms a 2D liquid phase in which the proteins have high in-plane mobility (Fig. 3a, $t_0$ + 18.92 s). Gradually, the proteins become visible as small domains of two to four coaligned nanorods that are short-lived, often appearing for only a single frame (Fig. 3a, $t_0$ + 75.68 s to $t_0$ + 277.03 s), but, with time, become larger and longer lived (Fig. 3a, $t_0$ + 277.03 s to $t_0$ + 337.84 s) until a stable smectic phase emerges (Fig. 3a, $t_0$ + 337.84 s to $t_0$ + 940.54 s).

To quantify the dynamics of assembly and the degree of order, we used a computational workflow where an ensemble of deep convolutional neural networks was employed to achieve semantic segmentation, which was then used as input to a conventional algorithm for rod recognition (Fig. 3b and Supplementary Figs. 4–6 for computational workflow details and ref. 28. for details of machine learning code). This approach was used to establish the total coverage (Fig. 3c), as well as the orientation and center of mass of each protein from each frame of the movies, which were in turn used to obtain the nematic (Fig. 3d) and smectic (Fig. 3e) order parameters, respectively (for detailed calculation of order parameters see Methods under "order parameters"). As the analysis shows, virtually every nanorod visible in any frame is oriented along a single direction. Thus, the nematic order rises rapidly and saturates. In contrast, the smectic order increases gradually, growing slowly at first and then rapidly transitioning to higher values before saturating as the surface becomes densely packed (Fig. 3e). This behavior mirrors that of the surface coverage (Fig. 3c) and shows that, as the domains become closely spaced, a percolation threshold is reached at which the probability of rod attachment increases rapidly and leads to high coverage, while the high degree of translational mobility enables the domains to align and reach high smectic order.

In the case of f-mica at 3 M K⁺ (Fig. 4 and Supplementary Movie 2), rods are already visible in the initial frames, both as individual rods and small domains, and unlike the case of m-mica, are oriented along all

three K⁺ sublattice directions with roughly equal probability (Fig. 4a). Individual domains fluctuate in size but, on average, grow as individual rods attach (Supplementary Movie 2). Furthermore, unlike the m-mica case, no rapid transition in either coverage or order parameter is observed. Most importantly, the resulting HDD phase exhibits low values for the nematic order parameter, although this is expected to be the equilibrium phase at high surface coverage and mobility. Moreover, the smectic order parameter is always nearly zero.

The above results show that, although the proteins are designed to bind through an electrostatic interaction between the carboxyl side-chains of the glutamates and the K⁺ sites of the mica lattice, the resulting liquid crystal phases on m- and f-mica exhibit stark differences even though the K⁺ sublattices are identical. Moreover, the proteins exhibit much higher mobility on m-mica than on f-mica. Consequently, on f-mica the proteins remain trapped in an HDD phase even though the equilibrium state is nematic, while on m-mica they exceed nematic order, instead attaining a high degree of smectic order.

A source for the contrasting mobility and distinct liquid crystal phases is the distinct structure imposed upon the overlying solution by the underlying mica lattice. Although the proteins' direct interaction with the mica surface is through the three-fold symmetric K⁺ sublattice, they cannot avoid interacting with the surrounding solution. As in any colloidal system, solvent-exclusion forces arising from the Brownian motion of water can be expected to influence protein-protein and protein-surface interactions. In the case of m-mica, the broken symmetry of the near-surface hydration structure (Fig. 2c, e) could, in principle, impart a large enough bias in the orientational potential energy landscape to alter the alignment of the nanorods.

To investigate whether a change in the potential energy landscape due to the altered symmetry of the water structure in going from f- to m-mica can lead to the observed behavior of DHR10-mica18 as a consequence of purely colloidal forces, we performed 2D grand canonical Monte Carlo simulations of hard rods (i.e., non-overlapping high-aspect-ratio rectangles) freely depositing on a surface in two scenarios (Fig. 5): 1) the three lattice vectors that define the rod orientations have

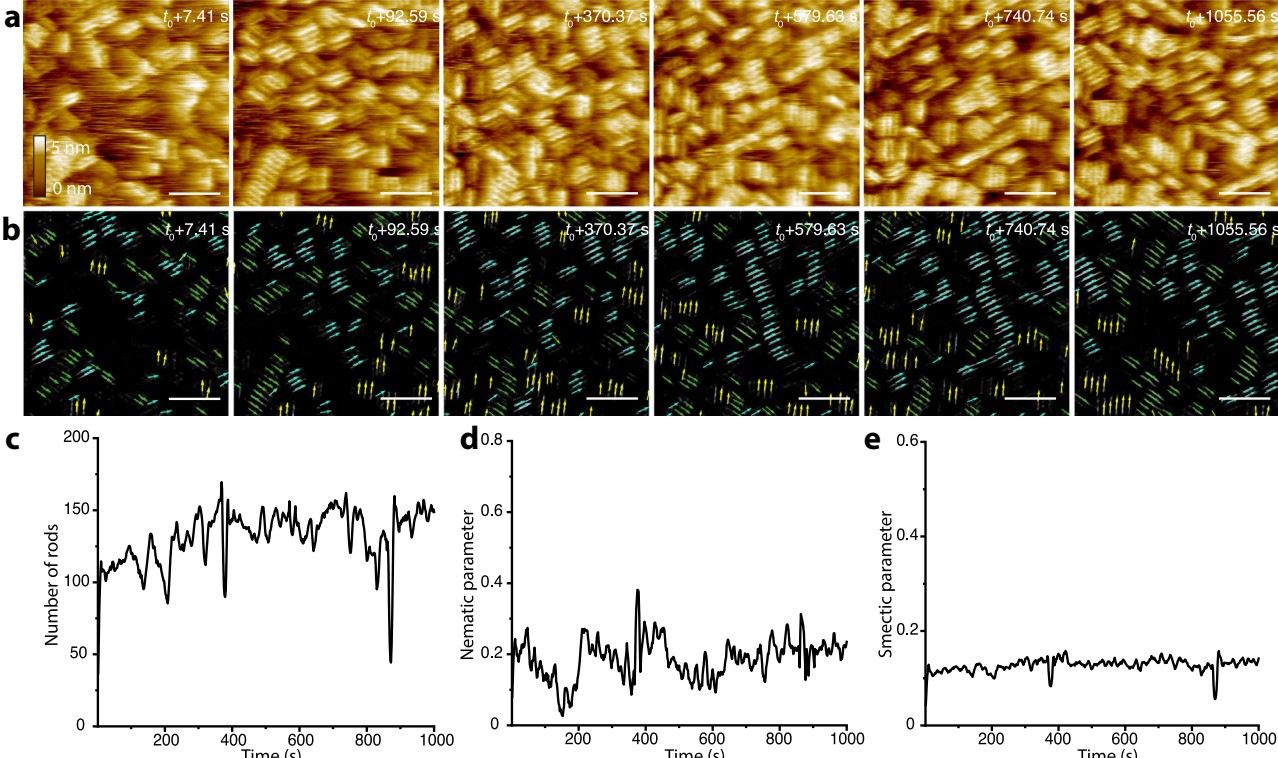

**Fig. 4 | In-situ high-speed AFM results and machine learning analysis to follow the assembly of protein nanorods on f-mica. a** HS-AFM images from Supplementary Movie 2 showing the translational and rotational motion of protein rods and their assembly on f-mica along the three K⁺ sublattice directions into a high-density disordered phase. Scale bar equals 50 nm. **b** Machine learning-based workflow to recognize the protein rods in the HS-AFM images in (**a**), where each arrow indicates the placement and alignment of a nanorod. **c** Number of recognized protein rods as a function of time. **d, e** Nematic and smectic order parameters as a function of time, respectively, for the observed rod assembly in (**a**). Source data are provided as a Source Data file.

equal probability of occupancy − i.e., the potential is three-fold symmetric − to represent the case of f-mica (Fig. 5a, g–j), and 2) one of the three orientations is twice as energetically favorable compared to the others − i.e., the potential is quasi-two-fold − to represent m-mica (Fig. 5b, c–f). For each, we systematically varied the chemical potential, which controls the surface concentration of rods, and rod mobility, which determines the maximum distance a rod is allowed to move during a Monte Carlo step, selects between kinetically trapped states at low mobility and equilibrium states at high mobility, and captures the increasing mobility of the proteins with increasing salt concentration.

At each Monte Carlo step, each rod is allowed an attempt to translate elsewhere in the simulation box where it does not overlap with any other rods. These translation attempts are followed by a fixed number of evaporation or deposition attempts, where new rods may only be deposited where they would not overlap with any existing rods. These evaporation and deposition moves also implicitly account for rod rotations: a rod with a different orientation may be deposited in the same location as a recently evaporated rod as long as there is space available (detailed simulation procedure is described in Methods).

The results show that when the potential is three-fold symmetric (Fig. 5a), a rod mobility of zero leads to a three-fold disordered phase regardless of the value of the chemical potential, consistent with previous findings for low mobility rods on a triangular lattice[29]. As the chemical potential increases across simulations of rods with nonzero mobility, a transient nematic phase emerges (Fig. 4i) before being replaced, at high chemical potential, by a phase composed of large, ordered domains. The smectic order parameter (see Methods for the detailed calculation of order parameters) is nearly zero for all conditions where all three rod orientations are equally favorable. These results are consistent with the experimental observations for f-mica,

on which the protein mobility is too low for even small domains to reorient into alignment with their neighbors (Supplementary Movie 2).

In contrast, when one of the rod orientations is more energetically favorable than the other two even by just 2x (Fig. 5b), a clear smectic order emerges for all conditions investigated, provided the rods have adequate mobility and sufficiently high chemical potential, while a three-fold HDD phase is observed only at zero mobility and high chemical potential. The fact that we only observe the smectic phase when we apply an orientational bias to the model system suggests that the smectic phase observed for non-interacting rectangular rods is solely due to the emergence of a two-fold rod-surface potential on m-mica due to the structure of the interfacial water layers[14–16]. This result demonstrates that the smectic order observed in these systems is mediated by the molecular details of solution-surface interactions rather than rod-rod interactions[15,20].

Overall, the simulation results are consistent with the experimental observations of protein nanorod assembly on m-mica and f-mica at low and high ion concentration, respectively (Fig. 1c–f), with one caveat: while the simulation model predicts a three-fold HDD phase at high chemical potential for m-mica (Fig. 5d), it is only observed when the rods have zero mobility, a condition that currently cannot be corroborated experimentally.

## Discussion
The above findings show that designed protein-substrate interactions between proteins and substrates do not solely govern the 2D assembly of proteins on surfaces. Rather, the entropic forces associated with solvent-colloid interactions must be accounted for. However, the results also show that these colloidal forces differ from those in the bulk solution because the substrate's symmetry is imprinted on the solvent structure near the interface. Most importantly, as shown by

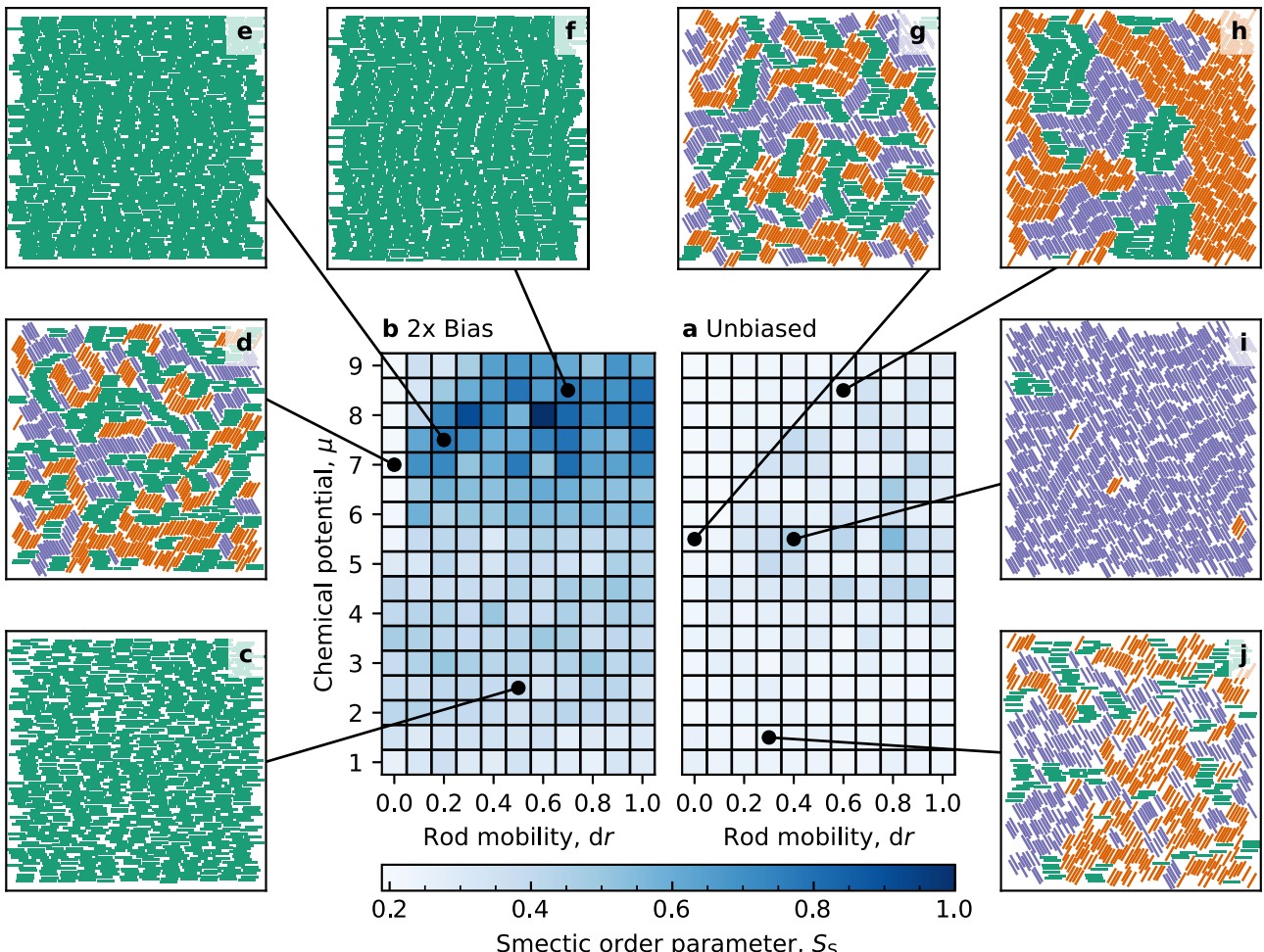

**Fig. 5 | Grand-canonical Monte Carlo simulations of hard rods on surface with and without bias. a, b** The smectic order parameters for a collection of Monte Carlo simulations of hard rods with aspect ratio $\ell = 7$ are presented in grids, where each box in the grid represents a single simulation defined by its chemical potential and rod mobility; the coloring indicates the value of the smectic order parameter. Separate grids are plotted for simulation collections where (**a**) all three rod orientations are equally favorable ("Unbiased"), and (**b**) where the horizontal rods are twice as energetically favorable as the other orientations ("2x Bias"). **c–j** Several select simulations are annotated with a snapshot of the final rod configuration; in the snapshots, rods of the same orientation share the same color. Source data are provided as a Source Data file.

DHR10-mica18 protein nanorods on m-mica at 3 M K$^+$, the combination of designed interactions to create a preference for binding along the three equivalent directions of the K$^+$ mica sublattice and the two-fold bias introduced into the potential energy landscape by the interfacial solvent structure in response to the broken three-fold symmetry of the m-mica lattice enables the proteins to assemble into a smectic phase, which would otherwise not exist.

The results presented here also suggest a molecular approach to complement computational platforms for de novo protein design, such as Rosetta, to improve the design of protein assemblies on inorganic surfaces. While protein design has proven capable of designing specific patterns of molecular interactions both between proteins and substrates, as well as between the proteins themselves, incorporating the effect of the interfacial structure requires a modified approach because, unlike the case of proteins, there is no large database of interfacial structures or of protein behavior in that environment. A possible approach is as follows: For each new crystal surface, the hydration structure is determined either by molecular dynamics or 3D AFM or a combination of both. The protein design is then coarse grained to capture the important molecular features of the protein, such as the protein side chain chemistry, that are responsible for the interactions at the interface, essentially creating a rigid body with a pattern of interacting sites whose binding energies are specified from a set of potentials

of mean force. Computational codes that can capture both the solvent exclusion effects associated with shape complementarity[30] which already exist, and that can be modified to incorporate the non-uniformity of the interfacial hydration structure, which have not yet been developed, would then be used to predict the true outcome of assembly. If it diverges from the design target, this divergence can be turned into a loss function for a neural network that then modifies the design and the code is run again, eventually converging on a design that is predicted to reach the target outcome of assembly.

## Methods
### HS-AFM
The protein stock solution (detailed protein design and purification protocol is discussed by Pyles et al.[1]) was diluted to the desired concentration with an incubation buffer. The incubation buffer contained 20 mM Tris-HCl (Sigma-Aldrich 1 M, pH 6.5) and 3 M KCl (Sigma-Aldrich, 99%). Then, 10 μL of diluted protein solution was dropped onto freshly cleaved substrates, muscovite mica (Ted Pella) and fluorophlogopite (SPI Supplies) at room temperature and the imaging was started immediately to capture the assembly process in situ. Note, substrates used were attached on a specially designed pillar for high-speed AFM imaging (Oxford Instruments, Asylum Research). Video-rate scanning mode of Cypher ES Environmental

AFM (Oxford Instruments, Asylum Research) was used. Ultra-high frequency cantilever was used for imaging (Arrow™ UHFAuD and USC-F1.2-k.15, Nanoworld).

## Machine learning workflow for HS-AFM data analysis

The experimental data was denoised using AtomAI[28,31,32] to obtain a semantically segmented image. This image was then used as the input for instance segmentation using methods described in ref. 32 and shown in Supplementary Fig. 4. The locations of rod centers obtained in this manner were then used to calculate the order of the assembled rods as described below.

## Order parameters

We quantify nematic order for a given system configuration as the maximum eigenvalue of the matrix $N$,

$$
N = \begin{bmatrix} \frac{1}{n_{\text{rods}}} \sum_{i}^{n_{\text{rods}}} 2(\cos\theta_i)^2 - 1 & \frac{1}{n_{\text{rods}}} \sum_{i}^{n_{\text{rods}}} 2\sin\theta_i \cos\theta_i \\ \frac{1}{n_{\text{rods}}} \sum_{i}^{n_{\text{rods}}} 2\sin\theta_i \cos\theta_i & \frac{1}{n_{\text{rods}}} \sum_{i}^{n_{\text{rods}}} 2(\sin\theta_i)^2 - 1 \end{bmatrix} \quad (1)
$$

Where $q_i$ is the orientation of rod $i$, and $n_{\text{rods}}$ is the total number of rods in the configuration.

To calculate the discrete translational symmetry of the smectic arrangement of rods in 2D, we calculate the ratio between the peaks corresponding to the number of rods in the image and the longitudinal spacing between rows of rods in a Fourier transform (FFT). The center peak of the FFT corresponds to the total number of rods in the image.

For the simulation-based data, we compute a pixelated representation of the rod configuration ($500 \times 500$ pixels) by marking for each rod the single pixel that corresponds to the rod center. We then pad these single pixels so that each rod center is instead represented by a $5 \times 5$ block of pixels. Padding the rod centers in this way helps make the images of rod centers and the associated FFTs more visually clear and interpretable. The top row of Supplementary Fig. 5 shows the image resulting from this process for four different configurations. We then compute the Fourier transform of these pixelated representations (Supplementary Fig. 5, bottom), and normalize the FFT intensities so that the maximum intensity for each image is one. The maximum intensity in the FFT is in the center of the transform image and corresponds to the total number of rods in the original image. We then aim to extract the features from the FFT that correspond to the end-to-end spacings of the rods. For the simulation data, we achieve this by applying a peak-finding algorithm within three thin, ten-pixel-wide strips centered at the zero-frequency FFT rotated by 0°, 60°, and 120°. The second highest FFT intensity within these strips corresponds to this spacing and serves as the smectic order parameter (Supplementary Fig. 5, bottom row). This construction allows us to take advantage of the fact that, in the simulation data, rods are only aligned along 0°, 60°, and 120°, so we only need to search these directions in the FFT to find features corresponding to the end-to-end rod spacing between rows of rods. Moreover, this also allows us to assign smectic order parameters even to disordered configurations. While not perfect, constructing the order parameter in this way makes it possible to capture the translational order as well as rotational order, and subsequently distinguish smectic order from nematic order.

In the experimental data, rod centers were plotted for each frame of the experimental data videos, and an FFT image was generated for each plot. The FFT peak corresponding to end-to-end rod row spacing was found manually, as this location does not vary throughout the video. The intensity of the highest peak within a $20 \times 20$ pixel window around this peak was used to calculate the smectic order parameter (Supplementary Fig. 6). Looking within a window allows for finding the maximum value even if the peak location changes slightly within the experiment due to noise.

Defining two-dimensional smectic order as a ratio between the intensities of two FFT peaks is a start towards quantifying the order of liquid crystal-like films, but it is a simplification that discards information contained in the FFT, such as the width of the peaks, which could be used in the future to further define the order of two-dimensional smectic assemblies. Furthermore, this analysis does not consider the lateral spacing between rods, which did not vary in our experiments, because the data is normalized. But the FFT peak corresponding to lateral spacing might need to be considered for different systems.

## Simulations

We performed Monte Carlo simulations in the grand canonical ensemble, systematically exploring the combined effects of chemical potential, rod aspect ratio, rod mobility, and orientational bias. In particular, we performed a Monte Carlo simulation for each combination of: 1) Chemical potential, ranging from 1.0 to 9.0 in steps of 0.5. 2) Rod aspect ratio, equal to 3, 5, 7, or 10. 3) Rod mobility, ranging from 0.0 to 1.0 in steps of 0.1. 4) Orientational bias, where the horizontal rods are either equally energetically favorable or twice as energetically favorable as rods in the other two orientations.

We additionally perform a smaller subset of simulations with other orientational biases, making the horizontal rods 1.25, 1.5, 1.75, 3, or 4 times more energetically favorable than the rods in the other two orientations. This smaller subset includes simulations for rods with an aspect ratio $\ell = 7$ and for each combination of: 1) Chemical potential, ranging from 1.0 to 9.0 in steps of 1.0. 2) Rod mobility, ranging from 0.0 to 1.0 in steps of 0.2.

In all of the simulations we perform, we run the Monte Carlo algorithm for $10^7$ steps, recording data every $10^3$ steps and recording rod configurations ("snapshots") every $10^5$ steps. We use a simulation box size of $100 \times 100$ units with periodic boundary conditions and free placement of the rods (i.e., the rods are not restricted to a lattice). The baseline rod energy is set to -2 $k_B T$ so that, for example, rods that are 1.5 times more energetically favorable have an energy of -3 $k_B T$. The Monte Carlo code used was originally developed by Stephen Whitelam of Lawrence Berkely National Laboratory.

Each Monte Carlo simulation begins with an empty box. At each Monte Carlo step, an attempt is made to move each existing rod (if any are present in the box) by randomly picking a new proposed location for the rod. If, at the proposed location, the rod would overlap with any other rod, the translation is rejected; otherwise, the translation is accepted. After the rod translation attempts, 2000 attempts are made to deposit or evaporate a rod. In the case of deposition, a random location and orientation is chosen for the new rod. If this proposed location would have the rod overlap with any other existing rods, the deposition is rejected. Otherwise, the deposition is accepted according to the Boltzmann criterion, $\text{RAND}[0, 1) \leq \frac{A e^{\beta\mu - E}}{n+1}$, where $\text{RAND}[0, 1)$ indicates the choice of a random number between 0 (inclusive) and 1 (exclusive) from a uniform distribution, $A$ is the area of the simulation box, $\mu$ is the chemical potential, $E$ is the energy of the rod, and $n$ is the current number of rods on the surface. In the case of evaporation, a rod is selected at random and is removed from the box, also in accordance with the Boltzmann criterion $\text{RAND}[0, 1) \leq \frac{n}{A e^{\beta\mu - E}}$.

As a result of the energy balances in our Monte Carlo setup, orientational biases only begin to affect the rod configurations after the surface has reached full or near-full coverage. In other words, the Monte Carlo simulations typically first acquire a disordered configuration, which either remains throughout the simulation (based on the rod mobility) or anneals to a more ordered configuration, whether that be nematic, smectic, or having large, ordered domains.

A collection of figures showing the impact of changing the chemical potential, rod length, and orientational bias on the nematic order parameter are provided in Supplementary Figs. 7–9, 13, in addition to figures showing the relationship between the order parameters and

## Article

rod packing fraction (Supplementary Figs. 10, 11), an example illustration of the evolution of the smectic order parameter as the MC simulation progresses (Supplementary Fig. 12), and snapshots of the rod configurations at the end of the Monte Carlo simulations under various simulation conditions (Supplementary Figs. 14–26, Supplementary Data 1).

### Statistics and reproducibility

DHR10-micaN assembly on the two different micas has been repeated 3 times. The in-situ assembly processes were repeated on Cypher AFM and Bruker Nanoscope (for experimental details, see Methods).

### Reporting summary

Further information on research design is available in the Nature Portfolio Reporting Summary linked to this article.

## Data availability

The data that support the findings of this study are available from the corresponding authors upon request. High-resolution versions of Supplementary Figs. 14-26 are provided as Supplementary Data 1. Source data are provided with this paper.

## Code availability

The code used for the simulations is available from the corresponding authors upon request. The machine learning code used to analyze the AFM data is provided as Supplementary Data 2.

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

## Acknowledgements

We thank Prof. Deepak Dhar, INSA distinguished professor, Indian Institute of Science Education and Research (IISER), India, for the insightful discussion about theoretical models of rod assembly. This work was primarily supported by the Department of Energy (DOE), Office of Basic

Energy Sciences (BES), as part of the Energy Frontier Research Center program: CSSAS – The Center for the Science of Synthesis Across Scales, located at the University of Washington under Award Number DE-SC0019288 and at Pacific Northwest National Laboratory under contract FWP 72448. All AFM measurements, machine learning, data analysis, and simulations were conducted at PNNL. Initial development of the machine learning workflow was performed at the University of Tennessee, Knoxville and at Oak Ridge National Laboratory's Center for Nanophase Materials Sciences (CNMS), a DOE Office of Science User Facility. Synthesis of DHR10-mica18 protein for experiments on f-mica was performed at UW, with the initial design and synthesis supported by the US DOE BES Biomolecular Materials Program, award number DE-SC0018940. Initial development and optimization of image analysis algorithms to extract order parameters were carried out at PNNL with support from the DOE BES Division of Materials Science and Engineering under FWP 77246. Development of Monte Carlo codes was performed at the Molecular Foundry at Lawrence Berkeley National Laboratory, supported by U.S. DOE BES under Contract No. DE-AC02–05CH11231. A.S. acknowledges the support of the National Science Foundation Graduate Research Fellowship under Grant No. DGE-1762114 and DGE-2140004. PNNL is a multi-program national laboratory operated for DOE by Battelle under Contract No. DE-AC05-76RL01830.

## Author contributions

S.Y.S., B.H., and A.S. contributed equally to this study. S.Y.S. and S. Z. collected AFM data. A.S. and B.L. led the data analysis using the machine learning workflow. B.H. and C.J.M. led the Monte Carlo simulation. H.P. and D.B. conceptualized and designed the protein. J.C., J.R.E., Z.P., O.D., and S.W. contributed to the development of the Monte Carlo simulation model. M.Z. and S.K. developed the machine learning workflow. C.J.M., S.Z., and J.J.D.Y. jointly supervised this work. S.Z. and J.J.D.Y. conceptualized this work. S.Y.S., B.H., A.S., B.L., H.P., C.J.M., S.Z., and J.J.D.Y. wrote the manuscript.

## Competing interests

The authors declare no competing interests.

## Additional information

[1]Physical and Computational Sciences Directorate, Pacific Northwest National Laboratory, Richland, WA, USA. [2]Department of Materials Science and Engineering, University of Washington, Seattle, WA, USA. [3]Molecular Engineering and Science Institute, University of Washington, Seattle, WA, USA. [4]Department of Biochemistry, University of Washington, Seattle, WA, USA. [5]Institute for Protein Design, University of Washington, Seattle, WA, USA. [6]Molecular Foundry, Lawrence Berkeley National Laboratory, California, USA. [7]Center for Nanophase Materials Sciences, Oak Ridge National Laboratory, Oak Ridge, Tennessee, USA. [8]Department of Chemical Engineering, University of Washington, Seattle, WA, USA. [9]Department of Materials Science and Engineering, University of Tennessee, Knoxville, TN, USA. [10]Howard Hughes Medical Institute, University of Washington, Seattle, WA, USA. [11]Present address: Institute of Materials and Interfaculty Bioengineering Institute, Ecole Polytechnique Fédérale de Lausanne (EPFL), Lausanne, Switzerland. [12]Present address: National Center of Competence in Research Bio-Inspired Materials, University of Fribourg, Chemin des Verdiers 4, Fribourg, Switzerland. [13]Present address: Mattson Technology, Fremont, CA, USA. [14]Present address: Department of Chemical and Biomolecular Engineering, Johns Hopkins University, Baltimore, USA. [15]Present address: CNR-ISPC, Via Biblioteca 4, Catania, Italy. [16]Present address: Lambic Therapeutics, CA San Diego, USA. [17]These authors contributed equally: Sakshi Yadav Schmid, Benjamin Helfrecht, Amy Stegmann. ✉e-mail: chris.mundy@pnnl.gov; shuai.zhang@pnnl.gov; james.deyoreo@pnnl.gov

