## [Transparent Peer Review file · Nature Communications]

Impact of Solvent Forces and Broken Symmetry on the Assembly of Designed Proteins at a Liquid-Solid Interface

Corresponding Author: Dr James De Yoreo

Version 0:

Reviewer comments:

Reviewer #1

(Remarks to the Author)

The authors have resolved my technical questions and therefore the paper is suitable for publication in a journal. The novelty/impact still seems relatively low, and there is not much in the way of explanation of the phenomenon. I leave it as an editorial decision as to whether the manuscript meets the impact standard of the journal.

(Remarks on code availability)

Reviewer #2

(Remarks to the Author)

The manuscript by Schmid et al. investigates the assembly mechanisms of computationally designed proteins at solid-liquid interfaces. It addresses the common discrepancy between the intended supramolecular structures of de novo proteins and their observed "off-target" assembly on inorganic substrates. The authors present a physical explanation for this phenomenon, concluding that interfacial solvent forces, which are dictated by the substrate's deep crystal symmetry, can dominate the assembly process. The work integrates experimental observation with computational modeling. The authors have revised the manuscript, based on the comments of two previous referees, providing an improved version of their work

The methodology centers on a de novo designed protein, DHR10-mica18, engineered with the Rosetta software to form nanorods. This protein was designed to present a specific array of 54 carboxylate groups for electrostatic binding to the potassium sublattice of mica. The authors then compare the protein's assembly dynamics on two different mica substrates: muscovite (m-mica) and fluorophlogopite (f-mica). While these substrates possess identical surface lattices, they differ in their underlying crystal symmetry (C2/c for m-mica, C2/m for f-mica). The primary experimental technique is high-speed atomic force microscopy (HS-AFM), which permits real-time visualization of the assembly process. The resultant AFM image sequences are analyzed using a machine-learning pipeline to automatically recognize and track the position and orientation of individual protein rods. This quantitative analysis allows for the calculation of nematic and smectic order parameters to characterize the collective phase behavior on each surface.

The experimental results show a clear divergence in assembly outcomes. On m-mica, the protein rods organize into an ordered smectic phase. On f-mica, the same proteins form a high-density but orientationally disordered phase. This observation is notable given that the protein was designed to interact equivalently with the surface lattice of both materials.

To explain this discrepancy, the authors employ Grand Canonical Monte Carlo (GCMC) simulations of hard rods. These simulations demonstrate that the experimentally observed smectic phase is not the expected ground state for the rod system unless an external directional bias is introduced. The authors posit that this bias originates from the substrate's influence on

the interfacial solvent. The broken symmetry of m-mica is shown to impose an anisotropic, "striped" potential energy landscape on the solvent layer, which in turn directs the protein assembly. The higher symmetry of f-mica does not produce this effect, resulting in a different, unbiased assembly pathway.

Although, the protein used in this study is a de novo construct, and its sequence is necessarily artificial, in fact the design was optimized for a specific, non-biological function: binding to an inorganic crystal lattice, I feel that a comparative bioinformatic analysis of the artificial sequence against natural protein databases is missing. Such an analysis could contribute substantially to the insight on how the design process really optimizes the artificial sequences. The study's contribution is the identification of substrate-mediated solvent forces as a critical factor in protein assembly, suggesting that future design efforts must account for these environmental effects beyond direct protein-surface interactions. It would be useful to mention how the missing solvent interactions could impact more generally the protein modelling.

(Remarks on code availability)

Reviewer #1:

Comment: The authors have resolved my technical questions and therefore the paper is suitable for publication in a journal. The novelty/impact still seems relatively low, and there is not much in the way of explanation of the phenomenon. I leave it as an editorial decision as to whether the manuscript meets the impact standard of the journal.

Response: We appreciate the reviewer's recognition that we have addressed the technical questions but are surprised that the reviewer finds that the novelty is low and that we lack an explanation for the observed phenomena. Aside from the technical accomplishment of using high-speed imaging and machine learning to extract the dynamics of the ensemble with single-molecule resolution, there are two main findings of novelty:

The first is the discovery of previously unknown physics: while non-interacting rods are not predicted to form a smectic phase in two dimensions, either when the potential is isotropic or when it is three-fold symmetric, but rather can only reach nematic order (as is observed in our system with f-mica), we show for the first time that with even a small bias in the potential along one direction, the system reaches smectic order. This outcome is not described in textbooks or papers on colloidal systems, and thus this finding is truly novel. This finding is not just a speculation based on the experimental observation of smectic order; it is predicted as well by our Monte Carlo simulations. In addition, we are able to pinpoint the source of the bias, which is the anisotropy of the hydration structure, an outcome that again is both seen experimentally and predicted by molecular simulations. If by a "lack of explanation" the reviewer is referring to a reason why this behavior emerges at high salt concentration, we have provided one: the well-known Kirkwood transition leads to a reduction in screening length as KCl concentrations exceed 1M. We have only presented this explanation as a hypothesis, because to prove it would require a new, technically challenging investigation using surface second harmonic generation, which is far beyond the scope of this study.

The second point of novelty is the implication of the work for protein design platforms. Current design platforms determine the protein sequence needed to interface with the surface of crystalline materials by considering the interactions between the amino acid side chains and the surface-exposed lattice of the crystal. The results of this investigation show that the interfacial solution structures must also be taken into account. This should not be trivialized into a statement that solvent exclusion forces must, of course, be accounted for when particles interact with surfaces, because it is the structuring of the interfacial solvent that is the key factor in selecting the resulting organization of the proteins, not just the fact that there is a solvent exclusion force. Thus, this is a novel outcome of the research.

We have not modified the manuscript to address this comment, because these points are summarized in the following locations in the current manuscript:

1) *Abstract*: "Using Monte Carlo simulations, we reproduce the observed phases and show that an observed smectic phase, previously thought to be unstable for non-interacting rods in two dimensions, emerges when crystal symmetry introduces a directional bias. The findings demonstrate the importance of incorporating solvent forces as modulated by the hydration structure inherent to interfacial systems when designing protein assemblies at liquid-crystal interfaces."

2) *Introduction*: “In particular, the emergence of a 2D smectic phase, which is unexpected based on long-standing theories of colloidal systems^{21,22} as well as simulations^{23,24}, is shown to arise naturally when symmetry breaking creates a two-fold bias in the potential energy landscape. The results suggest a method for integrating coarse-grain simulations of colloidal systems into *de novo* protein design platforms, enabling the accurate design of supramolecular protein architectures at inorganic interfaces.”

3) *Conclusion*: “However, the results also show that these colloidal forces are distinct from those acting in the bulk solution because the symmetry of the substrate is imprinted onto the solvent structure near the interface. Most importantly, as shown by DHR10-mica18 protein nanorods on m-mica at 3M K⁺, the combination of designed interactions to create a preference for binding along the three equivalent directions of the K⁺ mica sublattice and the two-fold bias introduced into the potential energy landscape by the interfacial solvent structure in response to the broken three-fold symmetry of the m-mica lattice enables the proteins to assemble into a smectic phase, which would otherwise not exist.”

“The question then arises as to why the smectic phase only emerges at K⁺ concentrations of order 1M or larger. We hypothesize that this is associated with the Kirkwood transition, which is the concentration at which the Debye-Hückel approximation is no longer valid due to ion-ion correlation effects and the screening length no longer decreases with increasing electrolyte concentration but, instead, begins to increase. For 1:1 electrolytes like KCl, this transition is predicted to occur at concentrations of about 1M.³² This hypothesis is also consistent with the increased mobility observed at molar concentrations of KCl as compared to 100 mM or less.”

Reviewer #2:

Comment #1: Although, the protein used in this study is a *de novo* construct, and its sequence is necessarily artificial, in fact the design was optimized for a specific, non-biological function: binding to an inorganic crystal lattice, I feel that a comparative bioinformatic analysis of the artificial sequence against natural protein databases is missing. Such an analysis could contribute substantially to the insight on how the design process really optimizes the artificial sequences.

Response: We are thankful for insightful suggestion by the reviewer. The protein sequence used in this study, DHR10-mica18, was queried against natural protein sequences using the National Center for Biotechnology Information (NCBI) Basic Local Alignment Search Tool (BLAST). The lone match was the crystal structure of DHR10 (5CWG), the *de novo* designed helical repeat protein which served as the scaffold for the design of DHR10-mica18, with which it shares 84% sequence identity over the aligned region. No homology was detected to any natural sequences in the database.

Comment#2: The study's contribution is the identification of substrate-mediated solvent forces as a critical factor in protein assembly, suggesting that future design efforts must account for these environmental effects beyond direct protein-surface interactions. It would be useful to mention how the missing solvent interactions could impact more generally the protein modelling.

Response: We very much appreciate the reviewer's point. Although we did include a somewhat perfunctory statement about utilizing coarse-grain models to include the effect of the interfacial solvent structure, we refrained from prescribing a specific procedure, because there are a number of possible approaches and the degree to which each would succeed is unclear without attempting

them. The biggest challenge is that de novo protein design is an AI approach made possible by the huge database of known protein structures. However, there is no such database of either protein-crystal interfaces or interfacial solution structures of crystal surfaces. The basic design approach, ignoring the solvent, can work because there is a huge database of known crystal structures, and one can impose the simple geometrical constraint that a repeating pattern of appropriately charged amino acid side chains be congruent with the pattern of cations or anions in the crystal plane representing the surface. Because interfacial solution structures are only known for a handful of crystal surfaces, incorporating them into a design platform is not straightforward. In essence, a molecular dynamics (MD) simulation or a 3D AFM experiment must first be performed to get a solution structure. Then, a coarse-grain simulation that can capture the important molecular features of the protein that are responsible for the interactions at the interface must be carried out. Such codes exist, but, to the best of our knowledge, no one has ever introduced a *structured* solvent. Learning how to do this correctly will take some research. Once accomplished, a design protocol might go as follows: a design is first determined for the protein in contact with a crystal surface without considering the solution. The design is then coarse grained and its assembly at the crystal surface is simulated using a hydration structure of the crystal surface determined independently either by MD or 3D AFM, thus giving a prediction of the true outcome of assembly. If it diverges from the design target, this divergence is converted into a loss function for a neural network, which then modifies the design. The code is run again, and the process is repeated until a design is achieved that is predicted to reach the target outcome of assembly.

To address this comment in the manuscript, we replaced the last sentence of the Conclusion section with an expanded discussion, as follows:

While protein design has proven capable of designing specific patterns of molecular interactions both between proteins and substrates, as well as between the proteins themselves, incorporating the effect of the interfacial structure requires a modified approach because, unlike the case of proteins, there is no large database of interfacial structures or of protein behavior in that environment. A possible approach is as follows: For each new crystal surface, the hydration structure is determined either by molecular dynamics or 3D AFM or a combination of both. The protein design is then coarse grained to capture the important molecular features of the protein that are responsible for the interactions at the interface, essentially creating a rigid body with a pattern of interacting sites whose binding energies are specified from a set of potentials of mean force. Computational codes that can capture both the solvent exclusion effects associated with shape complementarity³³ which already exist, and that can be modified to incorporate the nonuniformity of the interfacial hydration structure, which have not yet been developed, would then be used to predict the true outcome of assembly. If it diverges from the design target, this divergence can be turned into a loss function for a neural network that then modifies the design and the code is run again, eventually converging on a design that is predicted to reach the target outcome of assembly.